

# Comparative description and ossification patterns of *Dendropsophus labialis* (Peters, 1863) and *Scinax ruber* (Laurenti, 1758) (Anura: Hylidae)

Angélica Arenas-Rodríguez[1,*], Juan Francisco Rubiano Vargas[2] and Julio Mario Hoyos[1,*]

[1] Facultad de Ciencias, UNESIS (Unidad de Ecología y Sistemática), Pontifica Universidad Javeriana, Bogotá, Colombia
[2] Facultad de Ciencias, Universidad del Bosque, Bogotá, Colombia
[*] These authors contributed equally to this work.

## ABSTRACT

Although comparative studies of anuran ontogeny have provided new data on heterochrony in the life cycles of frogs, most of them have not included ossification sequences. Using differential staining techniques, we observe and describe differences and similarities of cranial and postcranial development in two hylid species, *Scinax ruber* (Scinaxinae) and *Dendropsophus labialis* (Hylinae), providing new data of ontogenetic studies in these Colombian species. We examined tadpoles raining from Gosner Stages 25 to 45. We found differences between species in the infrarostral and suprarostral cartilages, optic foramen, planum ethmoidale, and gill apparatus. In both species, the first elements to ossify were the atlas and transverse processes of the vertebral column and the parasphenoid. Both species exhibited suprascapular processes as described in other hylids. Although the hylids comprise a large group (over 700 species), postcranial ossification sequence is only known for 15 species. Therefore, the descriptions of the skeletal development and ossification sequences provided herein will be useful for future analyses of heterochrony in the group.

## INTRODUCTION

Comparative morphological descriptions for a specific group of frogs have provided useful systematic characters since 1960 (e.g., *Cannatella, 1999*; *Duellman, Marion & Hedges, 2016*). However, most studies of frog morphological characters focus on adults (*Faivovich, 2002*; *Faivovich et al., 2005*; *Maglia, Pugener & Mueller, 2007*; *Wiens et al., 2010*; *Pyron & Wiens, 2011*; *Yildirim & Kaya, 2014*; *Duellman, Marion & Hedges, 2016*), and tadpoles have often been overlooked (*Alcalde et al., 2011*). Of those comparative studies that examine tadpoles, most consider external morphological characters and skeletal characters are often neglected (*Fabrezi & Lavilla, 1992*; *Faivovich, 2002*; *Maglia, Pugener & Mueller, 2007*; *Hoyos et al., 2012*; *Yildirim & Kaya, 2014*). When skeletal features are considered, the chondrocranium is most often described, while the postcranium is frequently ignored

Corresponding authors
Angélica Arenas-Rodríguez, angelica.arenas@javeriana.edu.co
Julio Mario Hoyos, jmhoyos@javeriana.edu.co

(e.g., *Orton, 1953*; *Starrett, 1973*; *Wassersug, 1980*; *Wassersug & Heyer, 1988*; *Haas, 2003*). However, as with other groups, relatively few detailed comparative morphological studies of hylid tadpole skeletal development have been completed. Given the diversity and recent taxonomic rearrangements of the hylids (*Duellman, Marion & Hedges, 2016*; *Jungfer, 2017*) is important to amass as much comparative information about the group as possible. Thus, there continues to be a pressing need to conduct comprehensive comparative studies of hylids developmental morphology.

Interspecific variations in morphology help to clarify taxonomic groups within in the Hylidae. The family is predominantly distributed across the Neotropical region (*Frost, 2018*; *Duellman, Marion & Hedges, 2016*) and comprises 706 species grouped into seven subfamilies: Acridinae, Cophomantinae, Dendropsophinae, Hylinae, Lophyophylinae, Pseudinae, and Scinaxinae (*Faivovich et al., 2005*; *Wiens et al., 2010*; *Duellman, Marion & Hedges, 2016*; *Frost, 2018*). Ossification sequences are known for only 15 species, and only eight of those include the postcranial skeleton: *Acris blanchardi* (*Havens, 2010*; *Maglia, Pugener & Mueller, 2007*), *Boana lanciformis* (former *Hyla lanciformis, De Sá, 1988*), *Boana pulchella* (former *Hypsiboas pulchellus Hoyos et al., 2012*), *Dryophytes chrysoscelis* (former *Hyla chrysoscelis, Shearman & Maglia, 2014*), *Dryophytes versicolor* (former *Hyla chrysoscelis, Sheil et al., 2014*), *Hyla orientalis* (*Yildirim & Kaya, 2014*), *Osteopilus septentrionalis* (*Sheil et al., 2014*), *Pseudacris crucifer* (*Havens, 2010*).

Because identifying variations in developmental morphology and ossification sequence can lead to informative phylogenetic characters (*Weisbecker & Mitgutsch, 2010*; *Harrington, Harrison & Sheil, 2013*), we provide a detailed anatomical comparison of the cranial and postcranial development (including the sequence of onset of ossification) between two species of Andean hylids, *Dendropsophus labialis* and *Scinax ruber*.

## MATERIALS AND METHODS

We cleared and double stained for bone and cartilage. We made some changes to the standard protocol of *Dingerkus & Uhler (1977)*: (1) proportion of ethanol (SIGMA Ref. 459836-2L) and acetic acid (SIGMA Ref. K36101663 620) was changed to 70:30; (2) the Alcian blue (SIGMA Ref. A5268-25G) was increased to 75 mg/add volume, which was dissolved in ethanol and acetic acid; and (3) staining duration of this last solution was increased to 72 h. A number of specimens in our series of *S. ruber* did not show clear staining, thus we increased our sample size. The sample size for tadpoles and metamorphs of *D. labialis* was $N = 32$, and *S. ruber* was $N = 114$. The number in each series corresponds to the availability of specimens stored at the Museo de Historia Natural "Lorenzo Uribe" at the Universidad Javeriana (MUJ) and the Instituto de Ciencias Naturales at the Universidad Nacional in Bogotá—Colombia (ICN). The larval stages of *D. labialis* were collected from the Municipio Tenjo, Cundinamarca Departament, 3,200 m (MUJ 9250). The larval stages of *S. ruber* were collected from the Mun. Neiva, Huila Dep., 570 m; Mun. Granada, Meta Dep., 470 m (MUJ 3727, MUJ 6178, ICN 46015-46017). Tadpoles and metamorphs were staged according to *Gosner*'s (*1960*).

Observations and photographs were made with a stereomicroscope (Advanced Optical: Amersham, UK (stereoscopes), and Ottawa, Canada (cameras)) connected to a camera

**Table 1** Ossification sequence of cranial and postcranial elements in *D. labialis* (Peters, 1863).

| Rank | Gosner stage (Number of specimens) | Elements ossified | |
|------|------------------------------------|-------------------|--|
| | | Cranium | Postcranium |
| I | 26(3), 27(3), 28(2), 29(2), 31(3), 32(1), 33(1), 34(1) | | |
| | 35 (1) | | Transverse process I–V |
| II | 36 (2) | Parasphenoid | Transverse process VI–VIII |
| III | 37 (3) | Frontoparietal, exoccipital | Neural arches I–VIII |
| IV | 38 (1) | | Hypochord |
| V | 41 (3) | | Femur, tibiofibula, humerus, ilium, radioulna, clavicle, pubis, metatarsal III–V, coracoids |
| VI | 42 (3) | | Metacarpal IV, urostyle |
| VII | 45 (1) | Mentomeckelian, premaxilla, maxilla, angulosplenial, dentary | Manus IV proximal phalange, Metacarpal III and V, scapula, pedal digit IV proximal phalange, Metacarpal I and II, metatarsal I, prepollex |
| VIII | 46 (1) | Neopalatine, nasal, pterygoid, vomer, septomaxilla, squamosal | |

(Infinity 1; Lumenera Corporation, Ottawa, Canada) with white LED light and Image Pro Insight program (version 8.0.3). The drawings were made using a digitizing tablet (Bamboo Connect pen; Wacom, Portland, OR, USA) and edited using Adobe Illustrator 5. Anatomical nomenclature for tadpoles follows *Parker (1876)*, *Higgins (1921)*, *Jolie (1962)*, *Roček (1981)*, *Duellman & Trueb (1986)*, *Haas (1995)*; *Haas (1997)*, *Hall & Larsen (1998)*, *Maglia & Pugener (1998)*, *Cannatella (1999)*, *Haas (1999)*, *Sheil & Alamillo (2005)*, *Pugener & Maglia (2007)*, *Bowatte & Meegaskumbura (2011)*, *Hoyos et al. (2012)*, adult nomenclature is based on *Avilán & Hoyos (2006)*, using the Latin names given by the *ICVAN (1973)*.

The ossification sequence was determined by the first appearance of ossified bone. We refer to the metamorphic climax (MC) *sensu Banbury & Maglia (2006)* as the Gosner stages (GS) at which major modifications and fundamental structural changes occur, resulting in the loss of most of the larval characters. We also used the term "rank" to refer to the ordinal number within an ossification sequence at which an element begins to ossify. We note the first time any specimen at that stage showed stain. If two or more elements begin ossifying at the same time (Gosner stages), they were assigned the same rank (i.e., a tie) as per *Nunn & Smith (1998)*.

## RESULTS

Skeletal development and sequence of onset of ossification of the cranial and postcranial elements of *D. labialis* and *S. ruber* are showed in Tables 1 and 2. Many young specimens

**Table 2  Ossification sequence of the cranial and postcranial elements in *S. ruber* (Laurenti, 1758).**

| Rank | Gosner stage (Number of specimens) | Elements ossified | |
| --- | --- | --- | --- |
| | | Cranium | Postcranium |
| I | 26(3), 29(8), 30(7), 31(9), 32(8), 33(9),34(9), 35(14) | | |
| | 36 (10) | Parasphenoid | Transverse process I–VII |
| II | 37 (11) | | Neural arch I–III |
| III | 38 (4) | Frontoparietal, exoccipital | Transverse process VIII, neural arch IV–VIII |
| IV | 39–40 (3) | | Femur, tibiofibula, humerus, ilium, radioulna, scapula, hypochord |
| | 41(4), 42(5) | | |
| V | 43 (3) | | Ischium |
| | 44(4), 45(2) | | |
| VI | 46 (1) | Mentomeckelian, premaxilla, maxilla, angulosplenial, dentary, neopalatine, pterygoid, vomer, septomaxilla, squamosal | |

(between stages 26 and 35) stained poorly. However, we had sufficient sample size assess ossification sequence. Specimens older than stage 35 stained more reliably.

## Chondrocranium

We observed similar changes in the shape, size, and modification of structures in the development of chondrocranium in the two species. The elements of the skeleton were compared according to the initiation of ossification and not with a specific stage, because in the two study species the ossification occurred in different Gosner stages (Table 3). The overall width of the chondrocranium in *D. labialis* and *S. ruber* is roughly 80–90% of the total length (Fig. 1). The chondrocranium in *D. labialis* is wider (dorsal view) and lower (lateral view) than *S. ruber* (Figs. 1A–1C). Basicranial fenestrae did not differentiate with Alcian Blue in either species. We perceived a stronger blue coloration in *D. labialis*, and the jugular, prootic, and oculomotor foramen were clearly defined, whereas in *S. ruber* we could not see the oculomotor foramen.

The cartilaginous regions of the taenia tecti medialis and tectum sinoticum both represent a quarter of the basis cranii, extending from the frontoparietal fontanelle in both species. The tectum nasi roofs the nasal region, and the ethmoid plate forms the floor. The tectum nasi is separated from the orbit by a wall, the lamina orbitonasalis (=planum antorbitale *sensu Cannatella, 1999*). Because these regions are weakly chondrified, the lamina orbitonasalis is not observable in the tadpole stages, and the nasal capsules become visible after metamorphic climax (stage 42 and beyond). The taenia tecti marginalis is evident and clearly differentiated by GS37 in *D. labialis* and by GS35 in *S. ruber*. In neither

Arenas-Rodríguez et al. (2018), *PeerJ*, DOI 10.7717/peerj.4525

**Table 3  Onset of ossification of cranial and poscranial elements of *D. labialis* and *S. ruber*.**

| | Species | | Dendropsophus labialis | | | | | | | | | Scinax ruber | | | | | | | | | |
|---|---|---|---|---|---|---|---|---|---|---|---|---|---|---|---|---|---|---|---|---|---|
| | Gosner Stage (Specimens with ossified elements) | | 35 (1) | 36 (2) | 37 (3) | 38 (1) | 41 (3) | 42 (3) | 43 (1) | 45 (1) | 46 (1) | 36 (6) | 37 (9) | 38 (3) | 39/40 (3) | 41 (4) | 42 (5) | 43 (3) | 44 (4) | 45 (1) | 46 (1) |
| **Cranium** | | Paraesfenoids | | 1 | 1 | 1 | 3 | 3 | 1 | 1 | 1 | 3 | 5 | 2 | 3 | 3 | 3 | 3 | 3 | 1 | 1 |
| | | Exoccipital | | 1 | 1 | 1 | 3 | 3 | 1 | 1 | 1 | | | 3 | 3 | 3 | 4 | 3 | 3 | 1 | 1 |
| | | Frontopariental | | | 1 | | 2 | 3 | 1 | 1 | 1 | | | 2 | 2 | 2 | 1 | 3 | 3 | 1 | 1 |
| **Post cranium** | Transverse Process | I | | | 1 | | | | | | | 6 | 9 | 3 | 3 | 3 | 4 | 3 | 3 | 1 | 1 |
| | | II | 1 | 1 | 1 | 1 | 3 | 3 | 1 | 1 | 1 | 5 | 6 | 3 | 3 | 3 | 4 | 3 | 3 | 1 | 1 |
| | | III | 1 | 1 | 1 | 1 | 3 | 3 | 1 | 1 | 1 | 5 | 6 | 3 | 3 | 3 | 4 | 3 | 3 | 1 | 1 |
| | | IV | 1 | 1 | 1 | 1 | 3 | 3 | 1 | 1 | 1 | 5 | 6 | 3 | 3 | 3 | 4 | 3 | 3 | 1 | 1 |
| | | V | 1 | 1 | 1 | 1 | 3 | 3 | 1 | 1 | 1 | 3 | 6 | 3 | 3 | 3 | 4 | 3 | 3 | 1 | 1 |
| | | VI | 1 | 1 | 1 | 1 | 3 | 3 | 1 | 1 | 1 | 3 | 6 | 3 | 3 | 3 | 4 | 3 | 2 | 1 | 1 |
| | | VII | 1 | 1 | 1 | 1 | 3 | 3 | 1 | 1 | 1 | 1 | 4 | 2 | 3 | 3 | 3 | 3 | 2 | 1 | 1 |
| | | VIII | | 1 | 1 | 1 | 3 | 3 | 1 | 1 | 1 | 1 | 2 | 2 | 3 | 3 | 2 | 3 | 1 | | |
| | | IX | | | 1 | | | | | | | | | 1 | 3 | 1 | 1 | 3 | | | |
| | Neural arch | I | | 1 | 1 | 1 | 3 | 3 | 1 | 1 | | | | 3 | 3 | 3 | 4 | 3 | 3 | 1 | 1 |
| | | II | | | 1 | 1 | 3 | 3 | 1 | 1 | | | | 3 | 3 | 3 | 3 | 3 | 3 | 1 | 1 |
| | | III | | | 1 | | | | | | | | 2 | 3 | 3 | 3 | 3 | 3 | 3 | 1 | 1 |
| | | IV | | | 1 | | | | | | | | | 3 | 3 | 3 | 3 | 3 | 2 | 1 | 1 |
| | | V | | | 1 | | | | | | | | | 2 | 3 | 3 | 3 | 3 | 2 | | |
| | | VI | | | 1 | | | | | | | | | 2 | 3 | 3 | 1 | 3 | | | |
| | | VII | | | 1 | | | | | | | | | 1 | 3 | 2 | | 3 | | | |
| | | VIII | | | 1 | | | | | | | | | 1 | 3 | 1 | | 3 | | | |
| | | IX | | | | | | | | | | | | | | | | | | | |
| | Manus | Metacarpal I | | | 1 | | | | | 1 | | | | | | | | | | | |
| | | Metacarpal II | | | 1 | | | | | 1 | | | | | | | | | | | |
| | | Metacarpal III | | | 1 | | | | | 2 | 1 | | | | | | | | | | |
| | | Metacarpal IV | | | 1 | | | 1 | | 2 | 1 | | | | | | | | | | |
| | | Metacarpal V | | | 1 | | | | | 2 | 1 | | | | | | | | | | |
| | | Metatarsal distal tarsal IV | | | 1 | | | | | 2 | 1 | | | | | | | | | | |
| | | Metatarsal distal tarsal IV | | | 1 | | | | | 1 | | | | | | | | | | | |
| | | Prepollex | | | 1 | | | | | 1 | | | | | | | | | | | |
| | Pes | Metatarsal I | | | 1 | | | | | 1 | | | | | | | | | | | |
| | | Metatarsal II | | | 1 | | | 1 | | 1 | 1 | | | | | | | | | | |
| | | Metatarsal III | | | 1 | | 1 | 2 | | 1 | 1 | | | | | | | | | | |
| | | Metatarsal IV | | | 1 | | 1 | 2 | | 1 | 1 | | | | | | | | | | |
| | | Metatarsal V | | | 1 | | 1 | 2 | | 1 | 1 | | | | | | | | | | |
| | | Atlas | 1 | 1 | 1 | 1 | 3 | 3 | 1 | 1 | 1 | | | 3 | 3 | 3 | 4 | 3 | 3 | 1 | 1 |
| | | Femur | | | 1 | | 2 | 3 | 1 | 1 | 1 | | | | 3 | 1 | | 3 | 1 | | |

Arenas-Rodríguez et al. (2018), *PeerJ*, DOI 10.7717/peerj.4525

**Table 3** (*continued*)

| Species | | | | | | | | | | Dendropsophus labialis | | | | | | | | | Scinax ruber | | | | | | | |
|---|---|---|---|---|---|---|---|---|---|---|---|---|---|---|---|---|---|---|---|---|---|---|---|---|---|---|
| Gosner Stage (Specimens with ossified elements) | 35 (1) | 36 (2) | 37 (3) | 38 (1) | 41 (3) | 42 (3) | 43 (1) | 45 (1) | 46 (1) | 36 (6) | 37 (9) | 38 (3) | 39/40 (3) | 41 (4) | 42 (5) | 43 (3) | 44 (4) | 45 (1) | 46 (1) |
| Fibula | | | 1 | | 2 | 3 | 1 | 1 | 1 | | | | | | | | | | |
| Tibia | | | 1 | | 2 | 3 | 1 | 1 | 1 | | | | | | | | | | |
| Tibiofibula | | | 1 | | 2 | 3 | 1 | 1 | 1 | | | | 3 | 1 | 1 | 3 | 1 | 1 | |
| Scapula | | | 1 | | 2 | 2 | | 1 | 1 | | | | 3 | 1 | | 3 | | | |
| Suprascapula | | | 1 | | 2 | 2 | | 1 | 1 | | | | | | | | | | |
| Clavicle | | | 1 | | 2 | 2 | | 1 | 1 | | | | 3 | 1 | | 3 | | 1 | |
| Coracoids | | | 1 | | 2 | 2 | | 1 | 1 | | | | | | | | | | |
| Humerus | | | 1 | | 2 | 3 | 1 | 1 | 1 | | | | 1 | 1 | | 3 | | | |
| Ulna | | | 1 | | | | | | | | | | | | | | | | |
| Radioulna | | | 1 | | 2 | 3 | 1 | 1 | 1 | | | | 3 | 3 | | 3 | | | |

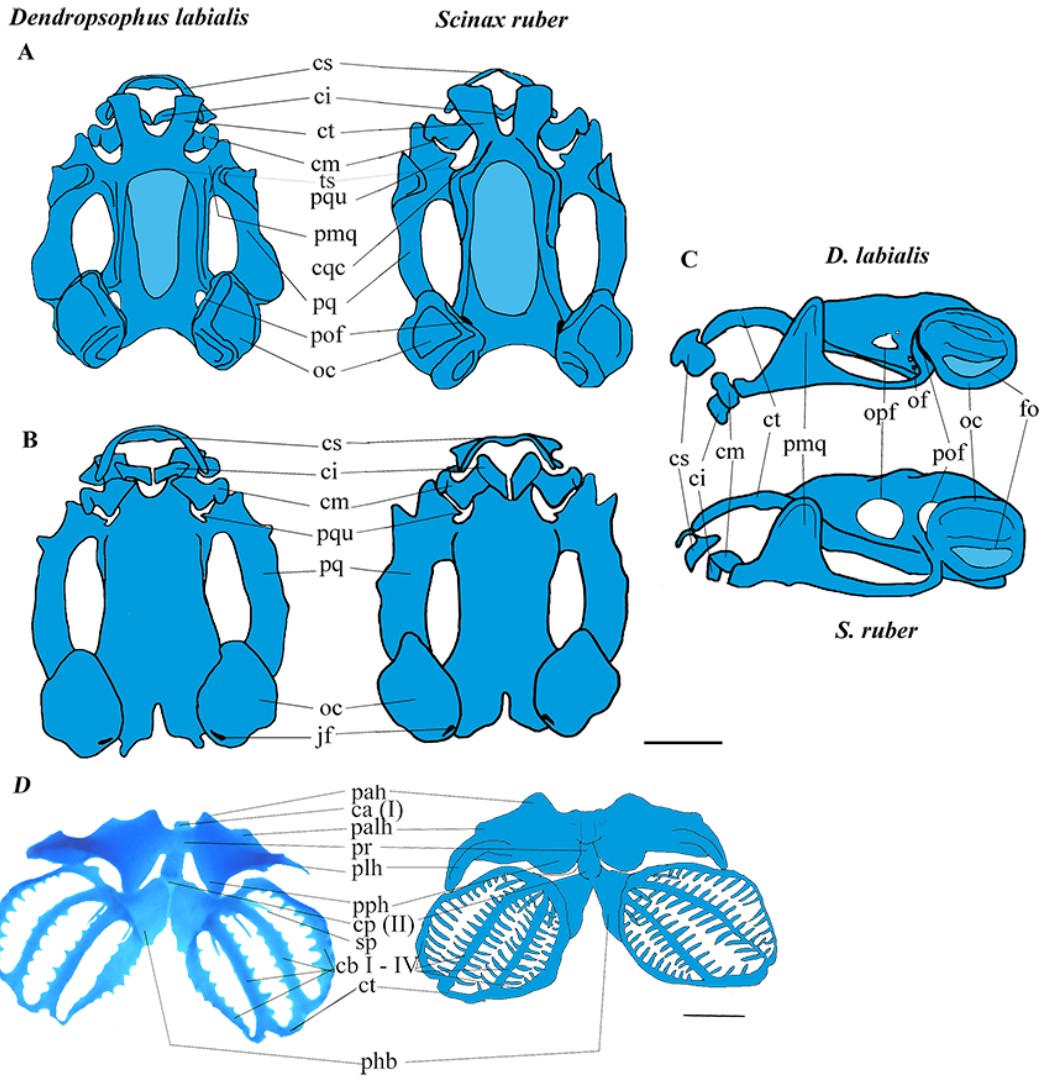

**Figure 1** **Larval chondrocranium of *D. labialis* (GS 34 - MUJ 9250) and *S. ruber* (GS 34 - MUJ 6178).**
(A) Dorsal view, (B) Ventral view, (C) Lateral view, (D) Ventral view of hyobranchial apparatus in
*D. labialis* (GS36 - MUJ 9250) and *S. ruber* (GS36 - MUJ 3727). Scale 1 mm. Chondrocranium: a, alae
suprarostralis; ci, cartilago infrarostralis; cm, cartilago Meckeli; cqc, commissura quadratocranialis;
cs, suprarostral cartilage; ct, cornu trabeculae; fo, fenestra ovalis; jf, jugular foramen; pal, processus
anterolateralis; pmq, processus muscularis quadrati; pof, prootic foramen; pq, palatoquadrate; oc, otic
capsule; of, oculomotor foramen; opf, optic foramen, ts, tectum sinoticum. Hyobranchial apparatus:
ca (I), copula anterioris; cb I–IV; ceratobranchialis I–IV; cp (II), copula posterioris; ct; commissura
terminalis; pah, processus anterioris hyalis; palh, processus anteriolateralis hyalis; phb, planum
hypobranchiale; plh, processus lateralis hyalis; pph, processus posterioris hyalis; pr, pars reuniens;
pqu, processus quadrato ethmoidale; sp, spicula. Blue: cartilage, light blue: fontanella.

species did we observe a frontoparietal fenestra, nor was a taenia tecti transversalis visible
on the edge of the frontoparietal fontanelle (Fig. 1A).

*Suprarostral cartilage.* In both species, the suprarostral cartilage is composed of a
discontinuous cartilaginous plate divided into a corpus suprarostralis and a pars alaris;

posterolaterally we observed a distal syndesmotic junction between the corpus and the ala. The ala has three processes: two rounded anterolateral processes that join syndesmotically with the cornu trabeculae, and one process posterolaterally (Fig. 1C). Fenestrations were not observed in the suprarostral cartilage, nor in the adrostral cartilage near the processus posterodorsalis (=processus dorsalis posterior, *sensu Bowatte & Meegaskumbura, 2011*). In *D. labialis* the corpus suprarostral is curved, while in *S. ruber* it is straighter and wider distally, articulating proximally with the cornu trabecula (trabecular horn, *sensu Cannatella, 1999*). The cornua trabecula are approximately 35% of the total length of chondrocranium (lateral view) in both species, but they are shorter and narrower in *D. labialis* than in *S. ruber*. The cornu trabeculae articulate anteriorly with the corpus rostrale and laterally with the pars alaris of the suprarostral cartilage.

*Cartilago Meckeli.* The cartilago Meckeli (=Meckel's cartilage, *sensu Cannatella, 1999*) has three processes: the retroarticular (short and blunt), the dorsomedial, and the ventromedial. These processes articulate with the infrarostral cartilage (commissura intramandibularis, *sensu Cannatella, 1999*) which is composed of two syndesmotically, joined flat plates, and the processus muscularis quadrati. The shape of the processus dorsomedialis and the processus ventromedialis are the same in both species. The palatoquadrate cartilage and the commissura quadratocranialis are joined anteriorly to the base cranii. Laterally, the palatoquadrate cartilage forms the arcus subocularis. The process muscularis quadrati is joined to the processus antorbitalis (=pars plana *sensu Parker, 1876*; = lamina externa *sensu Higgins, 1921*; = processus antorbitalis *sensu (Roček, 1981)*; = triangular plane *sensu Hall & Larsen, 1998* = cartilaginous planum triangulare *sensu Pugener & Maglia, 2007*) anterolaterally, projecting above the cornu trabecula. The processus hyoquadrati of the palatoquadrate cartilage articulates ventrally with the ceratohyalia of the hyobranchial apparatus (Fig. 1D).

*Otic capsule.* This structure is longer and higher than wide, occupying about a fifth of the total length of the skull. The crista parotica exhibits a more pronounced lateral projection in *D. labialis* than in *S. ruber*. The crista parotica is laterally developed, forming a small processus posterolateralis (=processus lateralis posterior *sensu Bowatte & Meegaskumbura, 2011*) and a small processus anterolateralis (more developed in *D. labialis*). The processus anterolateralis projects vertically, descending obliquely and overlapping the ventral posterolateral margin of the palatoquadrate cartilage. The otic capsule is perforated by the fenestra ovalis, which occupies about 20% of the otic capsule.

*Hyobranchial apparatus.* The large ceratohyal has a processus anterioris hyalis, a processus posterioris hyalis, and a processus anterolateralis hyalis. The first two processes are longer than the third, which extends to meet the transverse crease of the processus lateralis hyalis.

The basihyal plate is oval and extends proximally to the copula anterior (=Basibranchial I *sensu Duellman & Trueb, 1986*; = basihyale *sensu Haas, 1995*; *Haas, 1997*; = copula I *sensu Maglia & Pugener, 1998*; *Sheil & Alamillo, 2005*) in *D. labialis*, but is absent in *S. ruber*. The basibranchial plate is semi-oval and located between the two hypobranchial plates (=planum hypobranchiale *sensu Haas, 1999*; = plate hyoid *sensu Maglia & Pugener, 1998*; = hyobranchial plate *sensu Sheil & Alamillo, 2005*), and a branchial bridge is present

in both species, being wider in *S. ruber* than in *D. labialis*. The junction between each ceratobranchium and the planum hypobranchiale is syndesmotic. The ceratobranchia are united posteriorly by the commissura terminalis and bear three spicules anteriorly (Fig. 1D).

The chondrocranial morphology and hyobranchial apparatus is generally similar between the species examined here and those previously studied. However, we did identify several differences between *S. ruber* and *D. labialis*, including: (1) the shape of the suprarostral, (2) the size and width of infrarostral cartilages, (3) the length of processus articularis, (4) the thickness of palatoquadrate, (5) the size of optic foramen, (6) the presence of an operculum and processus posterolateralis of the otic capsule, (7) the thickness of the processus muscularis quadrati, (8) the attachment of the ascending process to the braincase, (9) the thickness of the planum ethmoidale, (10) the development of the branchial apparatus, (11) the presence of the copula I, and (12) the type of junction between the ceratobranchia and planum hypobranchiale (Figs. 1 and 2). These differences likely represent species specific differences between the two taxa examined.

## Appendicular skeleton

*Shoulder girdle.* The pectoral girdle is arciferal in both species. The earliest ossification of the clavicle, coracoid, and scapula appears at GS36 (Fig. 3A). The clavicle and the cleithrum are distinct, and an epicoracoid cartilage is prominent between the clavicle and the coracoid. The epicoracoids are not mineralized. In *D. labialis* the omosternum is elongated, and the sternum has two projections. The omosternum and the sternum are oval in *S. ruber*. The clavicle articulates with the coracoid, which is ossified in *D. labialis* at GS41 and in *S. ruber* at GS46. The sternum is formed by the epicoracoid and the mesosternum, which joins the medial junction of the epicoracoids (Fig. 2B).

*Pelvic girdle.* In both species, the primordium of the ilium appears at GS34 and is fully developed by GS41. The ilium begins to ossify by GS41 in *D. labialis* and by GS39/40 in *S. ruber* and articulates anteriorly with the ventral surface of the lateral margin of the sacral diapophyses by GS42. The iliac crest appears dorsally prominent. The primordia of the pubis and the ischium appear at GS36, and are synchondrotically fused by GS38 in both species. The sacral diapophyses is wider in *D. labialis* that in *S. ruber*. The pubis is completely fused by GS40. The pelvic girdle is completely ossified with the halves fused at the midline, extending anterodorsally forming an angle of 55° with the head of the femur by GS45 (Fig. 3).

*Fore limb and hind limb.* The first cartilaginous elements of the forelimbs (radius, ulna, and humerus) appear at GS32, and those of the hindlimbs at GS33 (femur, tibia, and fibula). The tibia and fibula are fused in *D. labialis* by GS41 and in *S. ruber* by GS38. We observed ossification of the radius and ulna in *D. labialis* (GS41) and *S. ruber* (post metamorphic). The radius and ulna are fused in both species. Primordia of the four carpal and five tarsal elements appear by GS33 and complete development by GS41.

The phalangeal carpal formula is 3-3-4-4 and the phalangeal tarsal formula is 3-3-4-5-4 in both species. Metacarpals are curved and phalanges are cylindrical, having a conical shape at the tip of the terminal phalanges. Digits IV (manus and pes) and V (pes) begin to ossify

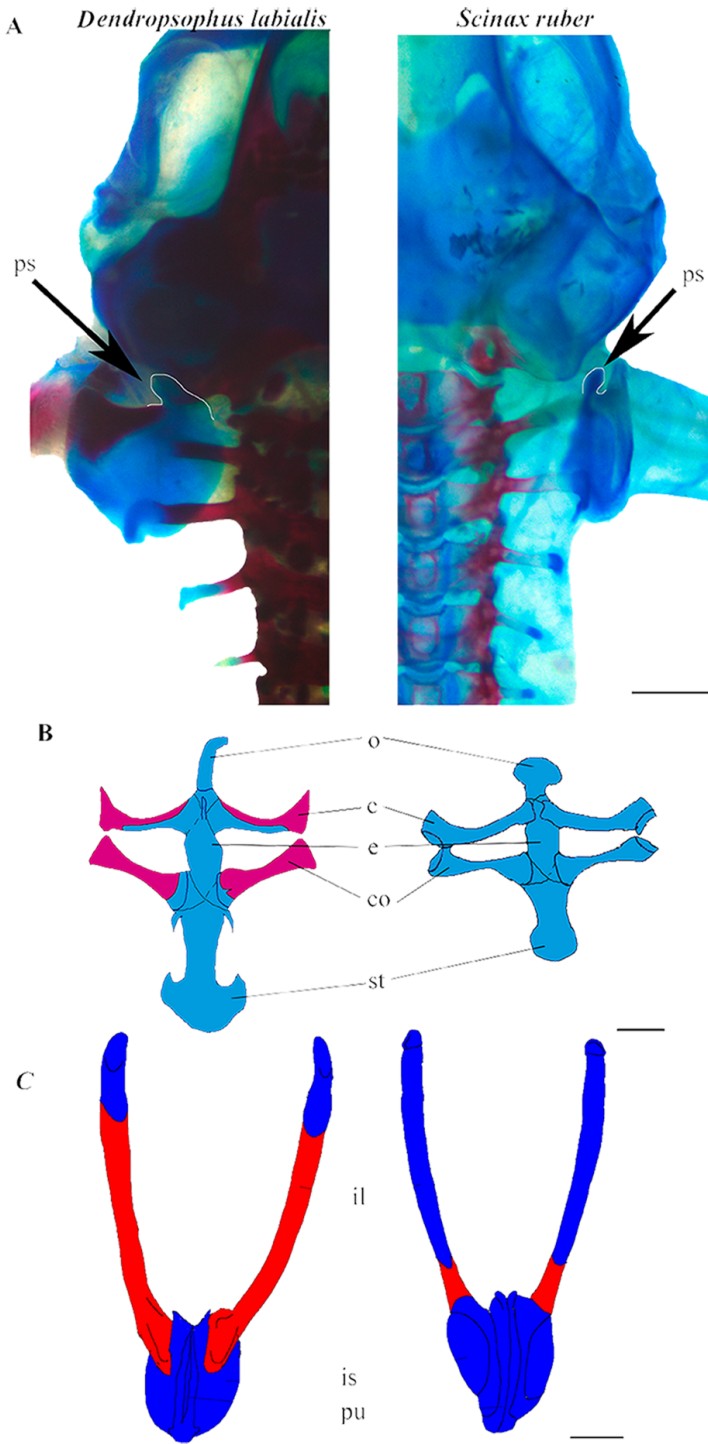

**Figure 2  Appendicular skeleton of *D. labialis* (GS45–MUJ497) and *S. ruber* (GS45–MUJ6018).** (A) Dorsal view of the scapula, (B) Ventral view of the pectoral girdle, (C) Ventral view of the pelvic girdle. Scale 1 mm. c, clavicle; co, coracoid; e, epicoracoid; ps, processus suprascapularis; o, omosternum; st, sternum; il, ilium; is, ischium; pu, pubis Red, ossified; blue, chondrified.

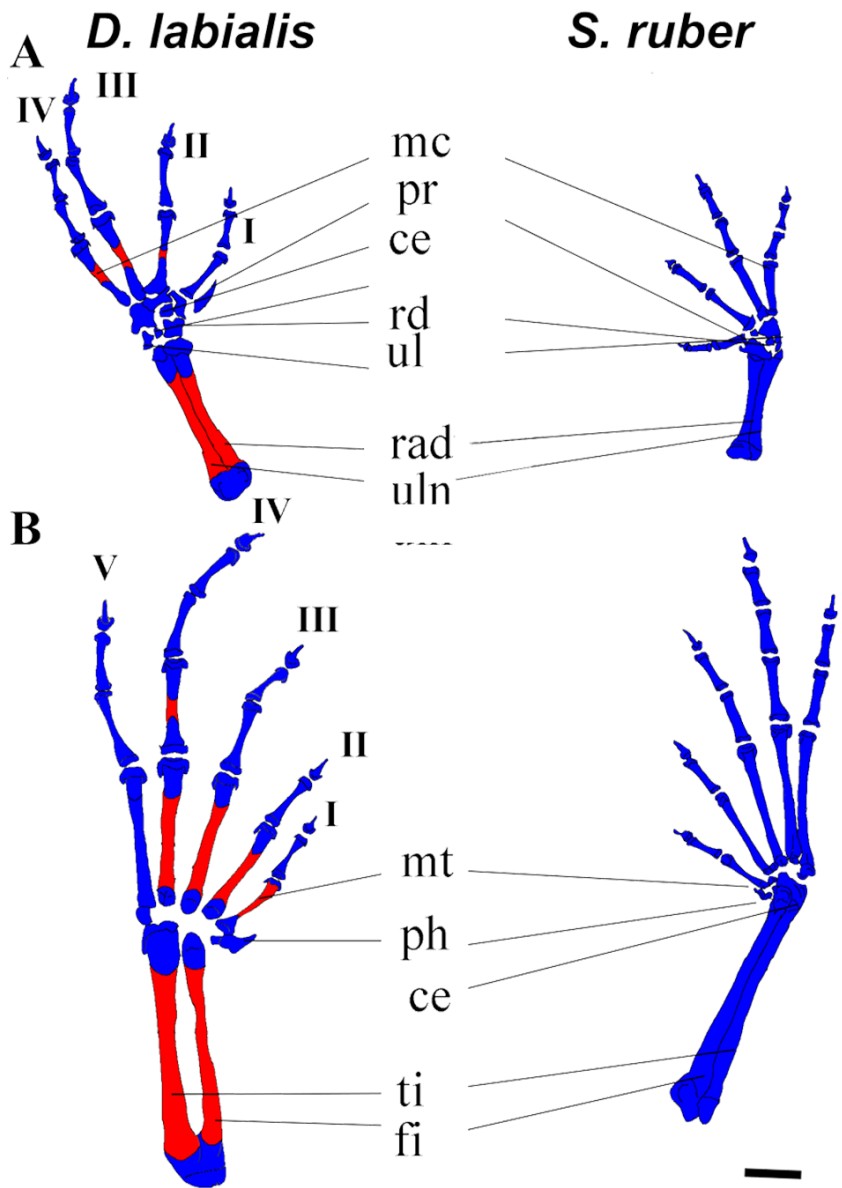

**Figure 3  Dorsal view of manus and pes of *D. labialis* (GS45–MUJ497) and *S. ruber* (GS45–MUJ6018).**
(A) Manus, (B) Pes. Scale 1 mm. ce, centrale; fi, fibulare; mc, metacarpal, mt, metatarsus; ph, prehallux; pr, prepollex; rd; radiale; rad, radioulna; ul, ulnare and intermedium; ti, tibiale. I–V, phalanges. Red, ossified; blue, chondrified.

by GS42 in *D. labialis*, although all phalanges are ossified at GS45 in both species (Fig. 3). The carpals were cartilaginous in all specimens and stages examined, and the distal tarsals were cartilaginous in *S. ruber*. The relative size of carpal elements is 3 < 4 < 2 < 1< prehallux and the tarsal elements is 4 < 5 < 3 < 2 < 1 < prepollex. Sesamoids are absent from GS25 to GS45. Figure 3A shows the limb elements (central, fibulare, radiale, tibiale, ulnare, and intermedium) at Stage 45.
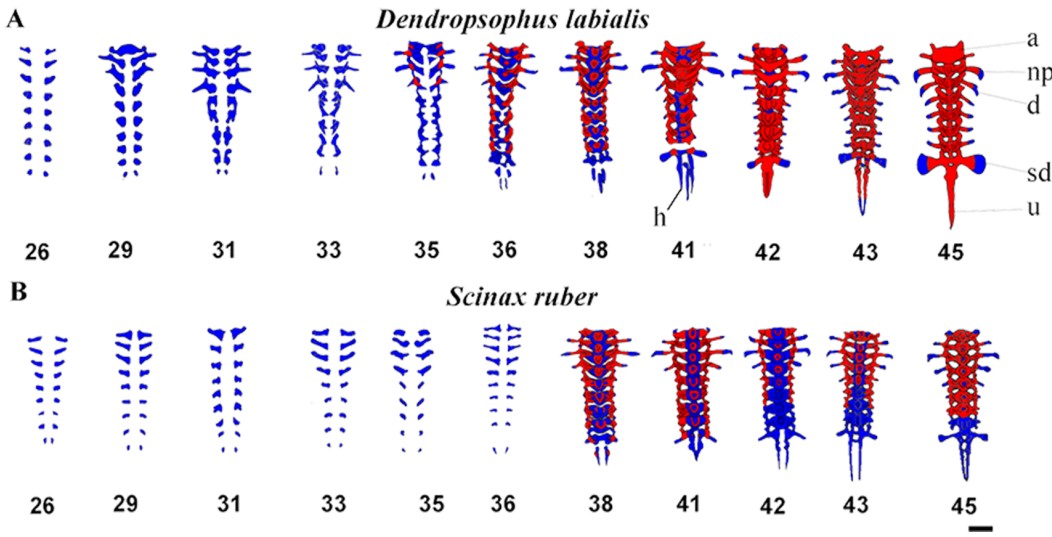

**Figure 4 Ventral view of ossification development in vertebral column of _D. labialis_ and _S. ruber_ at GS 26–45.** Scale 1 mm. h, hypochord; a, Atlas; np, neural process; d, diapophysis; sd, sacral diapophysis; u, urostyle Red, ossified; blue, chondrified.

## Axial skeleton

The vertebral column is composed of eight procoelous presacral vertebrae, the sacrum, and the urostyle. The notochord diminishes as the tadpoles grow and is completely resorbed by GS44 in both species (Fig. 4). We found that the axial skeleton was more chondrified in _D. labialis_ than in _S. ruber_. The first postcranial skeletal elements to develop in both species were the nine pairs of semicircular cartilaginous primordia of neural arches, including eight presacral vertebrae, the sacrum, the urostyle and the hypochord. The sacral diapophyseal primordia are cylindrical. The last postsacral vertebra (first coccygeal or Vertebra X _sensu Haas, 1999_) and the second coccygeal vertebra ossify only in _D. labialis_ by GS45. Simultaneous to the ossification of presacral vertebrae, there is notochord absorption, fusion of the coccygeal elements, and urostyle formation. The urostyle has a bicondylar articulation with the sacral vertebra and the condyles are widely separated in both species (Fig. 4).

The atlas is concave at its point of articulation with the convex occipital condyles at the base of the skull. Semicircular procoelous (_sensu Jolie, 1962_) vertebral centra begin to develop as early as GS31 in _D. labialis_ and GS32 in _S. ruber_, increasing the thickness of both the neural arches and the transverse process. The neural arches appear as cartilage at GS33 in both species, completing at GS34 in _D. labialis_ and at GS38 in _S. ruber_. The arches are fused dorsally at the midline at GS38 in _S. ruber_ and at GS38 in _D. labialis_. The transverse processes are the first elements to ossify in both species (Tables 1 and 2). Postzygapophyses and prezygapophyses are conspicuous in presacral vertebrae II, III, and IV in both species. Sesamoids are absent from GS25 to GS45.

## Ossification sequence

The earliest stage examined in both species was GS25. Ossification in *D. labialis* appears by GS34 and in *S. ruber* by GS35 (Figs. 1A and 4). Ossification in *D. labialis* begins with the atlas and the transverse processes, whereas in *S. ruber* it begins with the parasphenoid, the transverse processes I–VII and neural arches I–III.

The metamorphic climax (MC) begins at GS41 in *D. labialis* and GS39-40 in *S. ruber*. We identified seven ranks (I–VII) in *D. labialis* and five ranks (I–V) in *S. ruber* (Tables 1 and 2). Ossified elements were perceptible in *D. labialis* from GS35 to GS45, with 46 ossified elements, and from GS36 to GS43 in *S. ruber*, with 26 ossified elements. Metamorphic climax in *D. labialis* was at GS45 with 14 ossified elements and in *S. ruber* at GS39-40 with seven ossified elements. Of these, the structures in common are the femur, tibia, fibula, humerus, ilium, and radioulna.

## DISCUSSION

Despite Colombia housing the second greatest number of hylid species on the planet, few previous studies have considered developmental ossification of Colombian hylids. The family Hylidae has gone through a number of taxonomic rearrangements, as elucidated by various phylogenetic hypotheses based on molecular, chromosomal, and morphological data from both larvae and adults (*Faivovich, 2002*; *Faivovich et al., 2005*; *Wiens et al., 2010*; *Pyron & Wiens, 2011*; *Duellman, Marion & Hedges, 2016*). Data from additional morphological studies of Colombia hylids may help to support or refute these hypotheses.

Previous studies of the cranial morphology in hylid tadpoles include *Acris crepitans* (*Maglia, Pugener & Mueller, 2007*), *Boana lanciformis* (*De Sá, 1988*; *Alcalde & Rosset, 2003*), *Boana pulchella* (*Hoyos et al., 2012*), *Boana raniceps* and *Dendropsophus nanus* (former *Hyla raniceps* and *Hyla nana Fabrezi & Lavilla, 1992*; *Vera Candioti & Haas, 2004*, *Dryophytes versicolor* (former *Hyla versicolor*, *Sheil et al., 2014*), *Hyla orientalis* (*Yildirim & Kaya, 2014*), *Julianus acuminatus* (former *Scinax acuminatus*, *Fabrezi & Lavilla, 1992*; *Faivovich, 2002*; *Alcalde & Rosset, 2003*; *Alcalde et al., 2011*), *J. uruguayus, J. aff. pinimus* (former *Scinax uruguayus* and *Scinax aff. pinima*, *Alcalde et al., 2011*; *Rodrigues et al., 2017*), *Ololygon aromothyella* and *O. berthae* (former *Scinax berthae*, *Rodrigues et al., 2017*; *Alcalde et al., 2011*; *Faivovich, 2002*), *O. skuki* (*Rodrigues et al., 2017*); *Scinax granulatus* and *S. squalirostris* (*Rodrigues et al., 2017*; *Alcalde & Rosset, 2003*), *S. boulengeri* (*Rodrigues et al., 2017*; *Vera Candioti, 2007*), *S. fuscovariatus* (*Fabrezi & Vera, 1997*), *S. nasicus* (*Rodrigues et al., 2017*; *Vera Candioti, 2007*; *Vera Candioti, Lavilla & Echeverría, 2004*; *Fabrezi & Vera, 1997*), *S. ruber* (*Haas, 1996*). For a complete overview of the findings of these studies see Appendix 1.

Several of the differences between the two species examined here present interesting avenues for future examination. For example, the processus ethmoidalis of the quadrate in *S. ruber* is wide, and it is not clearly distinct from the processus articularis. In contrast, the processes of *D. labialis* are easily distinguishable and similar to that described by *Alcalde & Rosset (2003)*, who found similar features in *Boana raniceps* compared with the *Scinax* group (*S. squalirostris* and *S. granulatus, Scinax ruber* group). The palatoquadrate is similar

but the processus ascendens of the palatoquadrate in *D. labialis* is wider than in *S. ruber*, and the distal side of the cornu trabeculae extend posteriorly toward the otic capsule. The anterior region of the palatoquadrate is distinctively broader in *S. ruber* than in *D. labialis,* and in *S. ruber* the dorsomedial process is wider than the ventromedial process in *D. labialis*.

When comparing the development of *D. labilis* with *D. nanus* (*Vera Candioti, Lavilla & Echeverría, 2004* and *Alcalde & Rosset, 2003*), we found that *D. labialis* can be differentiated by the reduction of the buccopharyngeal and branchial basket structures and presence of processus quadrato etmoidale in ventral view. On the other hand, the information that is available for *Scinax* species (*Fabrezi & Lavilla, 1992*; *Haas, 1996*; *Fabrezi & Vera, 1997*; *Faivovich, 2002*; *Alcalde & Rosset, 2003*; *Vera Candioti, Lavilla & Echeverría, 2004*; *Vera Candioti, 2007*; *Alcalde et al., 2011*; *Rodrigues et al., 2017*) reveal that there are a numerous variations that requires extending the morphological studies in tadpoles.

*Scinax ruber* presents alae and corpus of suprarostral cartilage with deeper notches. The chondrocranium, hyobranchial apparatus, and the suprarostral body are joined syndesmotically as found by *Vera Candioti (2007)* in microphagous tadpoles of *S. nasicus* and *S. boulengeri. Dendropsophus labialis*, as *Dendropsophus nanus*, shows a suprarostral cartilage with corpus and alae forming a continuous structure, which is evidently associated with a deviation of the macrophogous mechanisms described by *Alcalde & Rosset (2003)*.

The lateral development of the crista parotica is more prominent in *S. ruber* than in *D. labialis*. It is possible that some of the variations in the anatomical structures of the otic capsule are functionally related to perception of vocalizations (i.e., same species recognition) in adult stages, but experiments must be conducted to confirm the relationship of these anatomical structures with hearing physiological functions (*Ruggero & Temchin, 2002*; *Boistel et al., 2013*).

The chondrification of skull in *S. ruber* is faint when viewed laterally, and foramina are not clearly visualized. By contrast, in *D. labialis* much more blue coloration was observed. This could be due to abundant chondrification of these parts or the early developmental stages of this anatomical area, in which allowed differentiation of craniopalatine carotid foramina.

Although the sample size for the *D. labialis* is very small in comparison with *S. ruber*, *D. labialis* exhibited more ossified elements with stronger chondrification and less intraspecific variation, while *S. ruber* showed more intraspecific variation and less overall chondrification in the samples (Fig. 1). *D. labialis* presented uniformly stained (ossified) elements in all individuals (Table 3).

This variation between *S. ruber* and *D. labialis* could be caused by intrinsic factors that determine the timing of development or by extrinsic factors affecting osteogenesis (*Vera & Ponssa, 2014*). It may not be a coincidence that *S. ruber* is a generalist species and *D. labialis* is an endemic one (*Frost, 2018*).

*Haas (1996)* reported that the ceratohyalia II–IV are fused in *Scinax ruber* and *Megophrys montana nasuta*, characteristics that separate them from other species. We confirmed this observation in *S. ruber* but not in *D. labialis*. The ceratohyal in *D. labialis* has a process on the articular condyle that is not present in *S. ruber*. *Alcalde & Rosset (2003)* found this

process in both *S. granulatus* and *S. squalirostris*. Spicules I–III on the posterior margin of the hypobranchial plate are present in *D. labialis* and *S. ruber*, but spicule IV is not.

Copula II is present in both species. Copula I is present in *D. labialis* as in *S. squalirostris*, but absent in *S. ruber* as in *S. granulatus*, *Boana raniceps* (*Alcalde & Rosset, 2003*), and *Tlalocohyla smithii* (*Vera Candioti & Haas, 2004*). Although the presence of copula I is extremely variable in hylids and is shared by all non-hylids (*Vera Candioti & Haas, 2004*), a relationship between this structure and the ecological function that it performs (e.g., prey utilization) has not been identified.

Additional characteristics of the developmental morphology of these species could place them with other hylids that have been studied previously. For example, the urostyle of *D. labialis* and *S. ruber* forms a bicondylar articulation with the sacral vertebra, and the condyles are widely separated. The shoulder girdle of both species present differences in the shape of the omosternum and sternum at GS 45. *D. labialis* and *S. ruber* present suprascapular processes in tadpoles and adults similar to those in other hylids (*Hypsiboas lanciformis*, *De Sá, 1988*; *Boana pulchella*, *Hoyos et al., 2012*; *Pseudacris crucifer* and *Acris blanchardi*, *Havens, 2010* and *A. crepitans*, *Maglia, Pugener & Mueller, 2007*) but is absent in *Scinax catharinae* clade (*Faivovich et al., 2005*).

Variations of larval characters between *Scinax* and *Dendropsophus* have been included in several phylogenetic studies (*Fabrezi & Vera, 1997*; *Haas, 1996*; *Haas, 1999*; *Haas, 2003*; *Alcalde & Rosset, 2003*; *Vera Candioti, 2007*). In our study, the skeleton shows significant differences between the species *S. ruber* and *D. labialis*, the elements ossified in *S. ruber* exhibit more intraspecific variability than in *D. labialis* (see Table 3).

Regarding ossification sequence, the first bones ossified in the cranium were the exoccipital, the frontoparietal, and the parasphenoid by GS 36. *Haas (1999)* found that in *S. ruber* this occurred one stage later by GS37. Similar to those that *Haas (1999)* described for other hylids, the ossification of the vertebrae begins from the centra of the presacral vertebrae and continues ventrally along the notochord, forming osseous rings around the notochord in both species. We found that the ossification of the centra in both species we studied begins ventrally and proceeds dorsally. *Haas (1999)* recorded the transverse processes of presacral vertebrae II–III as the first to ossify, while we found that it was the ossification of neural arches proceeds from I to IX in *D. labialis* (GS37) and from I to III in *S. ruber* (GS36) (Fig. 4).

The detection of more intraspecific variability in *S. ruber* than in *D. labialis* could also be due to the presence of more intra-generic diversity in the *S. ruber* clade. *Alcalde & Rosset (2003)* associated the type of feeding with the development of the lateral anterohyal process of the ceratohyal, between species with macrophage larvae (*Dendropsophus nanus*), and scraping microphages (*Boana pulchellus, Julianus acuminatus* and *S. nasicus*). This may indicate that morphological characteristics of the jaw may be involved in the particular feeding behaviors of these tadpoles, and therefore these traits would help to distinguish the species (Appendix 1).

Differences between the ossification sequences of these two species are also evident when examining the ossification ranks and number of ossified bones. In particular, *D. labialis* has more ranks in the sequence and more elements that begin ossification prior

to metamorphosis. With respect to the postcranium, the number of elements ossified appears earlier in *D. labialis* than in *S. ruber.* Because Gosner stages are based on external characteristics that rely on underlying skeletal change, it is only a relative measure of timing and should not be used as a way to compare species. Instead, we compared the relative timing of events in the ossification sequence by examining the order of onset of ossification of each element. *Nunn & Smith (1998)*, on page 86, considered ''ontogeny may be ordered by age, size, or stage; none of these measures are useful for comparing ontogeny across significantly divergent taxa''.

Table 4 outlines the ossification sequences of different species of the family Hylidae. The number of ranks that include elements of the skull and postcranium vary from one to five. The number of ranks increases when postcranial elements are included. *Weisbecker & Mitgutsch (2010)*, *Harrington, Harrison & Sheil (2013)*, and *Sheil et al. (2014)* used similar ranked ossification sequence data to reconstruct phylogenetic trees of amphibians in the families of Leptodactylidae, Ranidae, and Bufonidae. These researchers suggested using cranium and postcranium data, relating them to the type of development, and to include sequences of fossils, as far as possible.

Although the morphology and systematics of amphibians have been extensively studied (*Cannatella & Trueb, 1988*; *De Sá & Hillis, 1990*; *Báez & Pugener, 2003*; *Roelants & Bossuyt, 2005*; *Faivovich et al., 2005*; *Frost et al., 2006*; *Pyron & Wiens, 2011*; *Duellman, Marion & Hedges, 2016*), additional comprehensive descriptions of skeletal development and ossification sequences are needed to truly understand patterns of heterochrony in the group. Some of the biological implications of heterochrony, which are well known in amphibians (*Alberch, 1985*; *Reilly, Wiley & Meinhardt, 1991*), include changes in structure, and changes in the rate of growth of entire organisms (*Raff, 1996*; *Smith, 2001*; *Smith, 2002*; *Smith, 2003*). Some scholars have recognized that heterochrony may work as modules of developmental events with evolutionary implications that can promote or restrict the development of individual morphologies (*Wagner, 1996*).

Studies that have used statistical methods (e.g., Parsimov) to analyze ossification sequences have revealed heterochrony in the timing of onset of ossification in some cranial elements such as parasphenoid and prootic in *S. ruber* vs. *H. pulchellus* (*Hoyos et al., 2012*), or the frontoparietal, dentary, and maxilla in *D. labialis* vs. *Pseudis platensis* (*Fabrezi & Goldberg, 2009*). In our study, we found that the parasphenoid was the first element to ossify in both *D. labialis* and *S. ruber*, and the exoccipital, frontoparietal, and prootic were the second elements to ossify in both species.

It is possible that the difference in cartilage formation between the two species examined herein is due to paracrine factors induced in cells that express the mesodermal transcription factors involved in the activation of genes specific to cartilage (*Gilbert, 2000*; *Kozhemyakina, Lassar & Zelzer, 2015*); however, we did not account for these factors. Additionally, the intraspecific variation in the ossified elements between these species could be linked to specific genes (*Raff, 1996*).

**Table 4  Ossification sequences of different species of the family Hylidae, including postcranial elements.**

| Subfamily | Species | Element | No. ranks | Ossification sequence | References |
|---|---|---|---|---|---|
| Acridinae | Acris blanchardi | C | 9 | ps [ex, fp, po] [pm, sm]ma, ns, vo [an, de][me, pt, qj, qu, sq] sp | Havens (2010) |
| | | C, P | 11 | ps, ve [ex, fe, fp, po][fi, hu, mt, pf, ra, sc, ul, tf, ti][cl, co, ct, il, mc, ph][is, pm, sm]ma, ns, vo, [an, de][me, pt, qj, qu, sq] sp | |
| | Hyliola regilla | C | 6 | ps, fp [ex, po] pv [ma, ns, pm, sm, sq] [ag, de, pt][cm, me, pa, qj, sp] | Gaudin (1973) |
| | Pseudacris crucifer | C | 8 | ps [ex, fp] [pf, po] [ma, pm, sm] ns [an, vo] [me, pt, qj, qu, sq]sp | Havens (2010) |
| | | C, P | 13 | ps ,ve, fe [hu, il, ra, su, ul] [ct, ex, fp, fr, sc, tf, tl] [cl, co, mt, pf, po] [mc, ph] is [ma, pm, sm] ns [an, vo] [me, pt, qj, qu, sq] sp | |
| | Pseudacris triseriata | C | 4 | [ex, fp, pm] [de, ma, ns, pt, qj, sq, vo][m, po][cm, ha, pa, ps, sp] | Stokely & List (1954) |
| Dendropsophinae | Dendropsophus labialis | C | 3 | ps [ex, fp, po] [an, de, ma, me, np, pm, sm, sq, vo] | **This study** |
| | | C, P | 8 | [ve] [ps] [ex, fp, po] [cl, co, fe, hu, il, mt, ru, tf] [mc, sc][an, de, ma, me, np, pm, sm, sq, vo] | |
| Cophomantinae | Boana lanciformis | C | 8 | fp, ps, ex, po [ pm, sm] [ns, ma][an, de, sq][sp, me, qj, vo, pa, pt] | De Sá (1988) |
| | | C, P | 9 | fp [ps, ve] ex [fr, cl, co, ct, fe, hu, mc, mt, po, sc, tf, tl] pf [pm, sm] [ns, ma][an, de, is, sq][ap, me, pa, ph, pt, qj, sp, vo] | |
| | Boana pulchella | C | 2 | [ex, fp, ps] [an, de, ma, pm, po, sq] | Hoyos et al. (2012) |
| | | C, P | 4 | [ex, fp, ps, ve][fe, hu, il, ru, sc, tf][cl, co, ct, hy, mc, mt, pf, ph][an, de, ma, pm, po, sq] | |
| Hylinae | Dryophytes chrysoscelis | C | 8 | ps [ex][fe][fp][sm, pm, po][ma][de, ns, an, sq][vo] | Shearman & Maglia (2014) |
| | | C, P | 10 | ps [ex, na, cn][fe][sc, cl, co, hu, ra, ul, il, ti, fi, tb,mc, mt][ph, pf][fp][sm, pm, po][ma, is][de, ns, an, sq][vo] | |
| | Dryophytes versicolor | C | 6 | ps [ex, fp] [ma, pm, po] [an, de, sq] [pa, pt, qj] | Sheil et al. (2014) |
| | | C, P | 7 | ps [cl, co, fe, fi, fr, hu, il, na, ra, sc, ti, tl, ul] [ex, fp, mc, ph, pf, ve] [ma, pm, po] [an, de, sq] [ns, me] [pa, pt, qj] | |
| | Hyla orientalis | C | 5 | ps [ex, po] fp [ns, sm] [an, de, ma, pm, pt, sq, vo] | Yildirim & Kaya (2014) |
| | | C, P | 6 | ps [ex, hu, na, po, ve][ct, fe, fr, il, mc, tf, tl, ru][fp, cl, co, mt, pf, ph][ns, sm][an, de, is, ma, pm, pt, sq, pu, vo] qj | |
| | Smilisca baudinii | C | 7 | fp [ex, ps, sm] [ag, de, ma, pm, sq] [ns, pt, qj] [me, pa, pv][cm, sp] po | Trueb (1966) |
| | | | | [ex, fp, ps, sm][ma, pm, sq][ns, pt][pa, qj, vo][cm, et] so, po | Gaudin (1973) |
| | Triprion petasatus | C | 6 | fp, ns [an, de, ex, ma, me, pa, pm, ps, pt, qj, sm, sq][cm, pv] sp, po | Trueb (1970) |

**Table 4** (*continued*)

| Subfamily | Species | Element | No. ranks | Ossification sequence | References |
|---|---|---|---|---|---|
| Lophyohylinae | *Osteopilus septentrionalis* | | 3 | fp, sm [ag, cm, de, et, ex, ma, me, ns, pa, pm, po, ps, pt, qj, sq, vo] | *Trueb (1966)* |
| | | C | 7 | fp, sm [an, de, ex, ma, ns, pm, ps, pt, pv, sq] pa, qj [me, po, sp] cm | *Trueb (1970)* |
| | | | 5 | [fp, sm][ag, de, ex, ns, ma, pm, ps, pt, pv, sq] pa, qj [po, sp] | *Gaudin (1973)* |
| | | C, P | 13 | ps, ve, fe [hu, il, ra, su, ul] [ct, ex, fp, fr, sc, tf, tl] [cl, co, mt, pf, po] [mc, ph] is [ma, pm, sm] ns [an, vo] [me, pt, qj, qu, sq] sp | *Sheil et al. (2014)* |
| Pseudinae | *Pseudis platensis* | C | 5 | [fp, ps] [ex, po] [ns, pm, sq] ma [de, pt, vo] | *Fabrezi & Goldberg (2009)* |
| Scinaxinae | *Scinax ruber* | C | 3 | ps, [fp, po] [an, de, ma, me, np, pc, pm, sm, sq, vo] | **This study** |
| | | C, P | 7 | ps, ve [ex, fp, po][fe, hu, il, ru, sc, tf, tl] is [an, de, ma, me, pm, pc] [np, sm, sq, vo] | |

**Notes.**

C, cranium; P, poscranium; ag, angular; an, angulosplenial; ap, plectral apparatus; cl, clavicle; cu, columella; co, coracoid; ct, cleithrum; de, dentary; et, ethmoid; ex, exoccipital; fe, femur; fi, fibula; fp, frontoparietalis; fr, fibulare; ha, hyoid apparatus; hu, humerus; hy, hypochord; il, ilium; is, ischium; ma, maxilla; mc, metacarpals; me, mentomeckelian; mt, metatarsals; na, neural arches; nc, neural center; np, neopalatine; ns, nasal; pa, palatine; pc, coronoid process; pr, presacral vertebrae; pf, phalanges of feet; ph, phalanges of manus; pm, premaxilla; po, prootic; ps, parasphenoid; pt, pterygoid; pv, pre vomer; qj, quadratojugal; qu, quadrate; ra, radius; ru, radioulna; sc, scapula; sm, septomaxilla; sp, sphenethmoid; sq, squamosal; su, suprascapula; ti, tibia; tf, tibiofibula; tl, tibiale; tp, transverse process; ul, ulna; ve, vertebra (including na, nc, pr, tp); vo, vomer; C, cranial elements; P, postcranial elements.
(Rank = absolute time of ossification of various structures simultaneously).

## CONCLUSIONS

The contribution of ontogenetic data (development and ossification sequences of skeletal structures) provides further information to help understand the interactions between ontogeny and phylogeny in morphological and ecological diversity of frogs. Ossification sequence data combined with evolutionary hypotheses may shed light on patterns of development to be used in future phylogenetic hypotheses. As *Larson, De Sá & Arrieta (2003)* suggested, "variation in chondrocranial morphology in larval anurans can be phylogenetically informative, even among closely related taxa".

## ACKNOWLEDGMENTS

We want to thank Timothy Sosa for help with the English translation. We appreciate the contributions of the reviewers.

### Funding

This work was supported by Pontificia Universidad Javeriana, Paläontologisches Institut und Museum, Universität Zürich (Phenotypic and developmental plasticity and sequence heterochrony in anurans ID: 00003918), and by the Departamento Administrativo de Ciencias, Tecnología e Innovación—COLCIENCIAS (Colombian government Institution—Convocatoria 511/2010). The funders had no role in study design, data collection and analysis, decision to publish, or preparation of the manuscript.

## Grant Disclosures

The following grant information was disclosed by the authors:

Pontificia Universidad Javeriana.

Paläontologisches Institut und Museum.

Universität Zürich: 00003918.

Departamento Administrativo de Ciencias, Tecnología e Innovación—COLCIENCIAS (Colombian government Institution—Convocatoria 511/2010).

## Competing Interests

The authors declare there are no competing interests.

## Author Contributions

- Angélica Arenas-Rodríguez conceived and designed the experiments, performed the experiments, analyzed the data, contributed reagents/materials/analysis tools, prepared figures and/or tables, authored or reviewed drafts of the paper, approved the final draft.
- Juan Francisco Rubiano Vargas conceived and designed the experiments, performed the experiments, approved the final draft.
- Julio Mario Hoyos conceived and designed the experiments, analyzed the data, contributed reagents/materials/analysis tools, authored or reviewed drafts of the paper, approved the final draft.

## Animal Ethics

The following information was supplied relating to ethical approvals (i.e., approving body and any reference numbers):

The samples are from the museum of Javeriana University, Bogotá Colombia.

## Data Availability

Specimens belong to the Museo de Historia Natural "Lorenzo Uribe" at the Universidad Javeriana (MUJ) and the Instituto de Ciencias Naturales at the Universidad Nacional in Bogotá—Colombia (ICN). *Dendropsophus labialis* (MUJ 9250) and *Scinax ruber* (MUJ 3727, MUJ 6178, ICN 46015-46017).

## Supplemental Information

Supplemental information for this article can be found online at http://dx.doi.org/10.7717/peerj.4525#supplemental-information.

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
