# Peer review of "Comparative description and ossification patterns of Dendropsophus labialis (Peters, 1863) and Scinax ruber (Laurenti, 1758) (Anura: Hylidae)"

_PeerJ, doi:10.7717/peerj.4525_

## Round 0.1 · original submission · Major Revisions

I have assembled three reviews. Two of them see a lot of merit in the study with respect to the reporting of new data and quality of anatomical descriptions. However, both describe a number of major issues that need to be addressed and thus suggest major revisions are needed. A third reviewer also provides critiques, but recommends rejection at this point. However, I don’t follow the concern about significance of the study as being a requisite for PeerJ as the mission of the journal is based on publication of quality of science and not impact. Therefore, if you are able to address the numerous concerns by the reviewers I will be happy to consider a substantially revised manuscript. Reviewer 1 identifies important previously published work that should be discussed and referenced in the manuscript. Both Reviewers 1 and 2 suggest the paper be reorganized to emphasize the comparative nature of the study and provide better clarity to the discussion. Reviewer 3 requests the reporting of additional data and improvement of many of the figures. All reviewers identify specific grammatical corrections and emphasize the need to revise the manuscript for clarity. Note that Reviewer 2 has attached a pdf with specific comments embedded.

Reviewer 1 ·

Basic reporting

A enormous and important bibliographic references are absent. In this sense, data about the condrochranium and ossification pattern in Scinaxinae could be found for: Scinax acuminatus (Fabrezi & Lavilla 1992; Alcalde et al., 2011); S. boulengeri (Vera Candioti 2007); S. fuscovarius (Fabrezi & Vera 1997); S. granulatus (Alcalde & Rosset 2003); S. nasicus (Fabrezi & Vera 1997; Vera Candioti et al. 2004; Vera Candioti 2007), S. squalirostris (Alcalde & Rosset 2003), S. skuki (Rodrigues et al., 2017); Julianus. aff. pinimus, and J. uruguayus (Alcalde et al. 2011); Ololygon aromothyella and O. berthae (Alcalde et al. 2011). In the same way, exist data about condrochranium in Dendropsophinae (i.e. Dendropsophus nanus; Fabrezi & Lavilla 1992; Vera Candioti et al., 2004). Despite most of these references absent in the manuscript, being crucial to provide a proper discussion, the authors should perform an exhaustive reading of the bibliography available.

Experimental design

Please describe the methodology in more detail. for example:
“tadpoles of Dendropsophus labials (N= 32), and Scinax ruber (N=114), and one adult of each species were double stained for bone and cartilage following Dangerous and Uhler (1977). Voucher are stored in …….”
In addition in the entire manuscript the authors should perform an update in the taxonomy of Hylidae following Duerman et al., 2016 or use the taxonomy of Faivovich et al., 2005, but not a mixture of both.

Validity of the findings

The discussion section should be focused to compare related species. In this sense, S. ruber should be compared with the information available for Scinax species (i.e. Alcalde and Rosset, 2003; Alcalde et al., 2011), and inferences (with caution by the absence of data in most taxa) with julianus and other Scinaxinae (like Ololygon), in the framework of Hylidae. In Dendropsophus, the authors omitted references useful to make comparisons (see above) and the result obtained should be discussed considering the complete framework of Hylidae. The similitudes observed between between taxa not closely related: Dendropsophus labials (Dendropsophinae) and Scinax ruber (Scinaxinae) could be consequence of common origin, but also could be consequence of homoplasy with independent origin. So the absence of data not allow a clear comparison, and the authors should be caution with such issue.

Additional comments

Some other minors correction are provide to the authors:

Introduction Section
Line 36: deleted Frost, 2006. add Frost 2017 and/or Duellman et al., 2016
Line 52: delete Colombian. The fact that these taxa are distributed in Colombia is not really important for the study of the phylogenetic relationship of the group.
Line 36-38: review Hylidae taxonomy, now is considered Phyllomedusidae as a full family, so Hylidae (sensu Duellman et al., 2016) now is composed by 705 species (Frost, 2017).
Materials and methods Section
Line 45: add potential informative characters
Line 74: replace & by and
Line 240: Add Duellman et al., 2016
Line 244: replace Hyla by Boana,
Line 245: delete Scinax species:. Replace Scinax uruguayus and S. aff. pinima by the actual genera Julianus (J. uruguayus, J. aff. pinimus). Scinax aromothyella and Scinax berthae by Ololygon aromothyella and O. berthae.
Line 247: Gastrotheca riobambae is not a Hylid!, belong to Hemiphractidae. replace Hypsiboas pulchellus by Boana pulchella.
Discussion section:
Line 235-236: Is not true!. The 125 species of hylids frogs present in Colombia is less than half of the number of species assigned to Hylidae for Brazil.
Line 235-262: Is not a real discussion, only repeat the results obtained and none discussion is provided. Please re write these paragraph with a proper discussion (or delete or move the paragraph to introduction and results).
Line 288: Check this reference Hass, 1996.
Line 292: H. microcephala???? replace by D. nanus.

Tables and figures.
FIGURE 1. Could the authors provide photograph of the condrochranium?, as was performed in 1D?. the drawings are really poorly detailed, and multiples structures could not be proper identified with those pictures (mainly in 1A, and 1B).

TABLE 4. Phyllomedusa valianti belongs to Phyllomedusidae not Hylidae!. Please check the table in order to update the taxonomy.

Reviewer 2 ·

Basic reporting

The work "Comparative description and ossification patterns of Dendropsophus labialis (Peters, 1863) and Scinax ruber (Laurenti, 1758)(Anura: Hylidae)" propose a comparative description of the skeletal development of two species of anurans, Scinax ruber and Dendropsophus labialis. Although it presents valuable unpublished data, in my opinion the work has problems of organization and presentation.
1) The authors should contextualize or justify why they study these species? What is the hypothesis? What is the importance / contribution of this work? That they are not studied species of anura from Colombia would not be enough for their publication in an international journal with a wide audience.
2) I suggest to present the results in a comparative way, and that the idea of the development of the described structures be clearly reflected:
Results:
The data of the tables 1 and 2 would be useful to see in a comparative way
- The authors study the development of the skeletal system of two species of anurans, but in the description is not recorded in which stages / moments of development the structures, ossifications, etc., appear. This is a serious problem since the main objective of the work is to describe the development, and such as the results are presented, this is not reflected. In the section "ossification sequence" there is a summary on this subject, but it is not useful to transmit to the reader what are the elements (each one) that appear before or later, and what would be the difference between both species studied in this aspect.

3) Discussion
Since the authors says: “The species of the family Hylidae show variation in the taxonomic arrangements, as it was showed by the variety of phylogenetic hypotheses based on molecular, chromosomal, and morphological data from both larvae and adults”. It would be very useful to organize your ms so that it is clear what would be the support that the development characters would potentially give to the composition of the phylogeny, taxonomy and evolution of the group.
There are other problems of closing the paragraphs and discussing the results with what was previously published that are marked in the pdf.

Experimental design

no comment

Validity of the findings

no comment

Additional comments

I recommend to re-organize yours data because that they are more profitable in multiple aspects

Annotated reviews are not available for download in order to protect the identity of reviewers who chose to remain anonymous.

·

Basic reporting

This manuscript provides a thorough description of the skeletal development of two hylid frog species. It is clearly organized, and the anatomical terminology is appropriate.
For anatomical terms that have alternative uses in the literature, references have been provided. The English could be improved. In addition to several typos and grammatical mistakes, there are some instances of awkward phrasing that make it a bit difficult to understand.

The first paragraph of the introduction does not seem to fit with the rest of the manuscript---the link between ossification sequence and potential phylogenetic characters is not well developed. A discussion about sequence variation or heterochrony might make a more compelling argument.

The figures and tables, in general, are appropriate for the type of study and the data have been presented in a way that allows for the observations to be repeated. I would like to see a full table of specimens examined (so that I can see how many specimens from each stage were examined). Some of this information can be gleaned from the table(s) and figures, but not easily. This is particularly important when discussing degree of chondrification---i.e., could a lack of stain really be a lack of cartilage or just one or two poorly stained specimens? Is it hard to tell if the number of specimens is not listed.

The figures could be much more detailed—for example, are the craniopalatine and carotid foramina in D. labialis visible? They aren’t shown in Figure 1. You describe them in text but do not show them in the illustration. Also, Figure 5. Really doesn’t provide much information and should probably be deleted (or given more treatment in text).

There are some papers listed in text that are not cited in the reference section (e.g., Shearman and Maglia, 2014)

Experimental design

This is a descriptive paper, so the experimental design is minimal. For the type of study, the methods and approach are appropriate.
It is not entirely clear to me why this paper was written as a comparison of the two species. If the main goal (as described in the introduction and conclusions) is to provide baseline information to be used later in comparisons of heterochrony within phylogenetic context and/or for phylogenetic character, why not just describe each species individually? What new information does comparing the two species provide? Do the developmental differences relate to adult morphological differences? Providing more justification for the comparative approach in the introduction would make the paper more interesting, and the significance and new knowledge gained by comparing them should be discussed in the conclusions.

The authors present a clearly defined research question, namely how does the skeletal development of these two species differ from one another. While in a larger context, the question is relevant and meaningful, the authors do not provide much in the way of a larger comparative context. That being said, publication of these data will enable other to use them for broader studies.

Validity of the findings

The anatomical descriptions seem straightforward and thorough without being redundant. The sample size for the D. labialis is very small, and it is not clear how much influence this might have over perceptions of interspecific variation. It would be good to address this.

Line 243: First paragraph of the discussion---this list is not fully consistent with the list in the introduction. You’ve listed papers in the intro that include chondrocranial descriptions, but then do not include them on this list.

Line 317: Regarding the ossification sequence, the first bones ossified in the cranium were the exoccipital, the frontoparietal and the parasphenoid at GS 36. I’m a bit confused---this doesn’t seem consistent with the tables. In fact, in Table 2, shouldn’t exocippital be listed at GS38?

There is little in terms of conclusion beyond the fact that the data might be useful in future studies. I think the paper might be more interesting if you used the conclusions to comment/speculate on why you think there might be differences among these two species. Just a thought.

Additional comments

In general, this paper provides a detailed anatomical comparison between two species, and the ossification sequence information may provide some additional data that could be useful for larger comparative studies conducted in the future (although a more robust sample of D. labialis would have been preferred). The morphological descriptions are well done, but the illustrations could be a bit more detailed.

---

## Round 0.2 · Major Revisions

I have assembled reviews from two of the previous reviewers. They both agree that the manuscript is substantially improved, but identify additional issues that need to be addressed. I classified the decision as "Major Revisions" in accordance with one of the reviewers, but fortunately the suggestions to address the issues are very specific. In fact, one of the reviews has provided a heavily annotated and edited manuscript for your benefit. Please, take all of these suggestions into consideration and revise carefully before resubmission. Please carefully describe how you have addressed their concerns, particularly with respect to figures. I look forward to receiving your revised manuscript.

Reviewer 1 ·

Basic reporting

After reviewing the manuscript entitled "Comparative description and ossification patterns of Dendropsophus labialis (Peters, 1863) and Scinax ruber (Laurenti, 1758)(Anura: Hylidae)", I have observed that it has been considerably improved. However, many issues must be addressed in more detail. The list of changes proposed are detailed below.

Experimental design

No comment

Validity of the findings

The validity of the data obtained is of great importance for the study of anuran development patterns. However, these data are not used in the best way, giving a poor discussion regarding the available literature about development.

Additional comments

ABSTRACT
Delete: of neotropical frogs (the lack of data of ossification pattern is common in all anurans groups, being not restricted to the neotropical region).
Is not clear if the differences described are between both taxa studied, between related species or in fact correspond to intraspecific differences. Please re-write the sentence.
INTRODUCTION
LINE 28: Add reference
LINE 31: Add reference
Line 34-40: i really don’t understand this paragraph. The authors propose that the morphology in the hylids has contributed to the group's systematics because they are many species?. I suggest delete the paragraph or re-write these.
Line 47-51: Please include in each species with ossification data the specific name with which it was described. For example: Boana pulchella (as Hypsiboas pulchellus, Hoyos et al., 2012).

MATHERIALS AND METHODS
LINE 64: replace “These specimens belong to the” by “Specimens examined are stored at”.

RESULTS
LINE 101: replace “in the two species” by “both studies species”.

DISCUSSION
The discussion section is not properly organized, still multiple paragraphs are only repetition of the results obtained and the comparison and discussion with the data available in the literature is poor, and need more work.

LINE 251: Add de Sá, 1988
LINE 251-252: delete Alcalde and Rosset 2003 from D. nanus reference.
LINE 253: delete “species: J.”
LINE 255: delete “Ololygon species”
LINE 257: delete “Scinax species”
LINE 262-271: are results, nothing is discussed…delete.
LINE 265-277: discuss the data obtained in relation to the data available in literature.
LINE 275-279: Move this paragraph to the end of the discussion section, is a conclusion more than a discussion.
LINE 283: replace Hyla by Boana
LINE 284: compare first the palatoquadrado between D. labialis and the others Dendropsophus with data available (i.e. D. nanus), the same structure of S. ruber with the others Scinax species, and later could perform some inference in the Hylidae context.
LINE 299: But Alcalde and Blotto (2006) report this characteristic in Limnomedusa macroglossa, is not an hylid frogs, so the differences could not be explained so lightly.
Line 318: add while in Boana raniceps….
Line 327-330: multiples hylids frogs (specially studied in Scinax by Faivovich, 2005) present a suprascapular process, review the literature.
Line 339: detailed the species transferred to Dendropsophus adding the proper reference.
Line 343: replace “species of Hylidae” by hylids.
Line 351: the taxa assigned to Litoria nannotis by Hass (2003) belongs to Ranoidea (Pelodryadidae). Again the differences or similitudes founded could not be homologues.

FIGURES
Despite the explanation given by the authors, I still consider that chondrocranium drawing is not the best technique to observe the anatomical characteristic of this structure. As the drawings were not updated, I leave the decision on the images to consideration of the editorial in chief.

·

Basic reporting

improved from previous review; see editorial suggestions in attached document

Experimental design

improved from last version

Validity of the findings

appropriate, although some of the discussion paragraphs are a bit confusing and I was not really sure what points they were trying to make (see in-text comments).

Additional comments

In general, the manuscript is much improved. I have made editorial and content suggestions in the attachment.

---

## Round 0.3 · Minor Revisions

Thank you for submitting your revised manuscript and addressing the reviewers' concerns. I think a much improved manuscript is the result. I view your manuscript is essentially ready to be accepted, however I have read through it closely and noted a few remaining typographic and grammatical errors or places where the wording could be made more clear. As accepted articles in PeerJ go directly to the production process, it is best to address these now before the proof stage. I have attached an annotated pdf with the corrections that I identified. This may not be an exhaustive list. Please read through the manuscript closely, including references. I look forward to your revisions.

---

## Round 0.4 · accepted · Accept

Thank you for addressing my editorial comments. I hope you view the review process has aided your manuscript. It is now accepted.